# The dependency structure of the financial multiplex network model: New evidence from the cross-correlation of idiosyncratic returns, volatility, and trading volume

Dariusz Siudak 🆔 *

Division of Economics and Finance, Institute of Management, Lodz University of Technology, Lodz, Poland

* dariusz.siudak@p.lodz.pl

## Abstract

This work describes the design of a novel financial multiplex network composed of three layers obtained by applying the MST-based cross-correlation network, using the data from 465 companies listed on the US market. The study employs a combined approach of complex multiplex networks, to examine the statistical properties of asset interdependence within the financial market. In addition, it performs an extensive analysis of both the similarities and the differences between this financial multiplex network, its individual layers, and the commonly studied stock return network. The results highlight the importance of the financial multiplex network, demonstrating that its network layers offer unique information within the multiplex dataset. Empirical analysis reveals dissimilarities between the financial multiplex network and the stock return monoplex network, indicating that the two networks provide distinct insights into the structure of the stock market. Furthermore, the financial multiplex network outperforms the singleplex network of stock returns because it has a structure that better determines the future Sharpe ratio. These findings add substantially to our understanding of the financial market system in which multiple types of relationship among financial assets play an important role.

## 1. Introduction

Complex networks have emerged as a robust modeling methodology, particularly adept at capturing the multidimensionality and nonlinear dynamics inherent in financial systems. The intricate interconnections and interdependencies within financial markets necessitate analytical frameworks capable of representing such complexity. Recent studies have demonstrated the effectiveness of complex network approaches in modeling the multifaceted nature of financial markets, providing insights into systemic risk and market behavior [1]. Additionally, the nonlinear interactions prevalent in financial systems have been effectively analyzed using complex network methodologies, offering a deeper understanding of market dynamics [2,3]. The complex network approach is a powerful analytical framework that has been used to study and understand a wide range of complex systems [4,5]. Recent studies have focused on the statistical and topological characteristics of financial networks using a complex network

**Data availability statement:** The data can be found in the research data repository: Siudak, Dariusz (2024), "The multiplex stock market network", Mendeley Data, V2, doi: 10.17632/kc5chp4tzd.2; https://data.mendeley.com/datasets/kc5chp4tzd/2.

**Funding:** The author(s) received no specific funding for this work.

**Competing interests:** The authors have declared that no competing interests exist.

approach [6]. This approach has been widely and effectively applied to the study of the interconnectedness on the stock market [7], and provides a convenient and suitable means to gain a deeper understanding of the structural and functional properties of complex financial systems [8–10].

All companies in a financial market are interconnected and form a linked network based on different quantities. The correlation between entities has a deep inner impact. Therefore, a well-known approach to building enterprise networks is to construct a cross-correlation network [11,12]. This approach involves measuring the cross-correlation between pairs of time series, which indicates the degree to which the two series are correlated. By constructing a network based on these cross-correlation values, researchers can identify patterns and relationships that may not be apparent from looking at the individual time series in isolation. Complex financial systems can be modeled using a multiplex network consisting of several network layers. This type of multilayered financial network has not been explored in depth. There are multiple types of relationships among financial assets. Therefore, it is crucial to analyze financial markets through the lens of multilayer networks rather than isolated monoplex networks, which identifies the research gap. This study provides the backbone for a novel approach, and assumes that the atomistic view of relationships in the financial market is no longer valid. For the purposes of this study, the multiplex network structure includes layers representing different time series-based asset relationships, namely, the firm-specific factor of the log-stock return, the volatility of the stock return, and the stock market turnover.

In recent years, there has been a growing interest in the use of multilayer networks to model complex financial systems [13–23]. A multilayer network is a type of network that consists of multiple layers, each representing a different aspect of the system under study. Multiplex networks are a subset of multilayer networks. A multiplex network is formed by a set of vertices interacting simultaneously in multiple network layers [24], in which nodes are connected by more than one type of edge and each type of link represents a separate network layer [25]. By modeling financial systems as multilayer networks, it becomes possible to capture the interdependencies and interactions between different components of the system in a more realistic and nuanced way. One of the key advantages of multilayer networks is their ability to capture the heterogeneity and complexity of financial systems, as it can be difficult to represent this using monoplex network models. Unlike traditional network models that represent the system as a single layer of nodes and edges, a multilayer network captures multiple interconnected layers, each representing a different aspect of the system. In other words, monoplex network theory is not sufficient to model and explain the complex multi-faceted relationships of the financial market.

Network models play a crucial role in portfolio optimization by providing insights into the structure and dynamics of asset relationships within complex financial systems. Network models allow for a detailed mapping of how asset returns are interconnected and illustrate the dependency between network topology and its function, particularly in the context of financial networks. The structure of a network can significantly influence risk evaluation and asset performance in portfolio construction [26–29]. Financial network models significantly enhance portfolio optimization by providing a framework to analyze the complex interrelationships between assets [30], facilitating the identification of high-performing investments [31], and improving the robustness of portfolio strategies through dynamic analysis and noise reduction. Network peripherality serves as an important indicator for identifying optimal assets [32,33]. The recent studies reveal that peripheral nodes in a network can potentially yield better portfolio performance compared to central nodes [11,34–36], where the risk-return ratio plays a pivotal role. Stocks located in the network's peripheral region exhibit a more diversified composition and demonstrate reduced vulnerability to irregular stock price

fluctuations during market volatility [8]. This insight allows for the selection of assets that are more profitable, well-diversified, and less risky. On the other hand, the superiority of portfolios based on peripheral stocks or centrality vertices is contingent upon the prevailing conditions of the stock market over a specified time horizon [37,38]. This indicates that this area has not been well explored. Nonetheless, empirical results of the study [39] demonstrated that network-based approaches yield better out-of-sample performance compared to traditional pairwise correlation methods.

The aim of establishing the financial multiplex network was to integrate three essential elements of asset connections within the financial market into a unified network structure. The first layer $l_1$ considers the stock return, which has been purged of the systematic risk premium driven by common market factors. The second layer $l_2$ addresses risk, while the third layer $l_3$ considers stock trading liquidity. The financial multiplex network elucidates the interconnectivity of financial assets by incorporating data pertaining to their idiosyncratic returns, risk profiles, and trading liquidity. Idiosyncratic return analysis plays a crucial role in understanding the unique characteristics and performance of individual assets, as demonstrated in the following studies [40–43]. El-Nader and Al-Halabi [42] observe a positive idiosyncratic premium return for the UK financial market. The advantage of analyzing idiosyncratic return networks over the stock return network lies in its ability to capture individual asset-specific characteristics and interactions. This granularity allows for a more nuanced understanding of the relationships between individual assets and their 0s to overall market behavior. Therefore, the idiosyncratic return network offers a superior representation compared to the conventional network, which relies on total stock return, as evidenced in the following studies [44–47]. The idiosyncratic return network offers a superior representation compared to conventional networks by providing a more refined and accurate depiction of the relationships between financial assets. This advantage arises from the idiosyncratic return network's ability to isolate the unique risk and return characteristics of each asset, effectively filtering out the influence of systematic factors such as market-wide or sectoral trends. By focusing on residual returns, the idiosyncratic return network eliminates systematic bias, reducing noise and enhancing the detection of meaningful, asset-specific connections. This results in a clearer and more precise understanding of the intrinsic dependencies between assets. Conventional networks, which incorporate both systematic and idiosyncratic risks, often overestimate correlations driven by external factors, thereby obscuring genuine interdependencies. By concentrating on asset-specific characteristics and avoiding distortions introduced by common systematic factors, the idiosyncratic return network enhances the practical applicability of network-based insights in financial decision-making. Therefore, the idiosyncratic return network represents a significant advancement in the study and application of financial network analysis, offering deeper insights into asset interdependencies. In addition, a network based on total return, compared to the idiosyncratic return network, replicates to a greater extent the information received from the volatility network. The exclusion of systematic risk for the first layer of the multiplex network is intended to diversify the three-layer network, in order to avoid the redundancy of information carried by the second layer (volatility network).

By integrating these multiple layers, the multiplex network enables a more comprehensive understanding of the structure of the system and can reveal hidden patterns and interdependencies that may not be apparent in single-layer models. In this context, multiplex networks offer a promising avenue for advancing our understanding of the complex interplay between financial entities, and the underlying factors that shape financial systems.

The paper proposes a novel financial network that is compiled using the concept of a multiplex network, meaning a network that incorporates multiple datasets of connections between assets. Three layers of the stock network are combined into one linked multiplex structure.

Various types of interactions among the same set of stocks are described by the three layers of the financial market system, and this fills the research gap. In this study, the topological properties of the financial multiplex network (FMN), its three layers, and, in addition, the commonly studied cross-correlation of the stock return network (SRN) are investigated. This work also conducts a comprehensive analysis of the similarities and differences between the multiplex network and its separate layers and the stock return network. Fig 1 illustrates the overall framework of the study. Specifically, the following research questions are examined:

1. Is there a significant difference between the FMN and the SRN?

2. Does each layer of the stock network offer unique information within the FMN?

3. Which network more accurately reflects the interconnections between financial assets?

All three layers of the multiplex dataset are constructed from cross-correlation matrices. By computing the correlation coefficient for all pairs of assets, one can obtain a fully connected network in which all the edges between each pair of stocks are presented. Such fully connected networks inherently contain a significant amount of noise. Therefore, it is essential to filter out the relevant information contained in the complex structure of the cross-correlation matrix among the stocks on the financial market. In this study, I use the minimal spanning tree (MST) approach to identify the most important connections within each network layer and to reduce the complexity. The MST approach is used because of its computational simplicity, robustness, and intuitive visualization. Furthermore, the MST approach offers an automatic procedure with no parameters to select, thus ensuring comparability between networks. It involves constructing a tree that spans all nodes in the network while minimizing the total length of the edges that connect them [33,48], sectors [49,50], markets, or indices [51,52], and in detecting changes in market structure and dynamics over time [38].

To the best of my knowledge, this is the first study on the creation of the multiplex network that combines three different types of relationship between the assets on the financial market, analyzes its statistical properties, and provides a multi-faceted comparison with the three layers and with a separate network of the cross-correlation of the log-return of the stock. The identification and quantification of interactions between network layers can be carried out employing such tools as link-overlapping indicators, similarity measures, and link correlations [17,53]. The results of this study, based on data from the US stock market, reveal the relevant dissimilarity between the multiplex network and the other networks in terms of their topological properties. Specifically, differences in network structure, connectivity, community structure, and disassortative behavior are observed. Furthermore, compared to the other networks (the layers and the SRN), only the multiplex network has the small-world network property. However, all the networks, namely the multiplex network, its three layers, and the

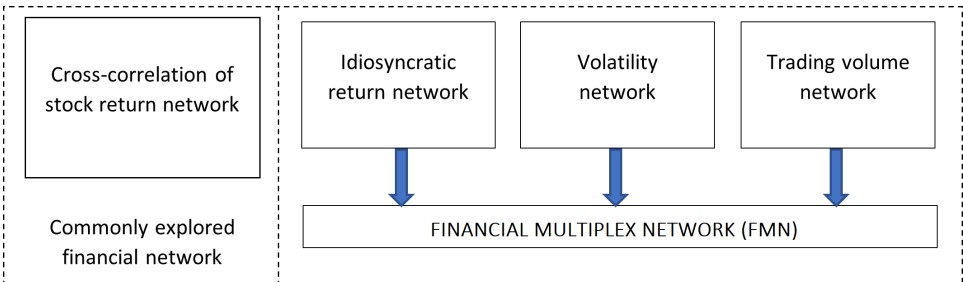

**Fig 1. General framework of the research study.**

stock return network, have in common that they obey the power-law vertex degree distribution. The results reported in this work provide evidence for the significance of the FMN in the sense that different network layers contribute distinct information to the multiplex dataset. In addition, the empirical analyses show that the FMN is different from the SRN, which means that the two networks take into account different information regarding the structure of the stock market. Moreover, it has been demonstrated that the FMN exhibits greater robustness to random node failures than the SRN, as the critical threshold for randomly removed vertices is considerably higher in the FMN.

An analysis of the effect of selected centrality measures of the two compared networks (FMN and SRN) on the out-of-sample Sharpe ratio was carried out. The findings revealed that a complex multiplex network structure achieves a higher dependency structure for the future stock performance under the risk–return relationship in the financial market than the monoplex network structure.

The main contribution of this research is twofold. First, in the area of complex networks, it involves the conceptualization of a novel financial network model, which provides a new perspective for the comprehensive analysis of stock price movements within the financial market. The multiplex dataset (FMN) encompasses a broader information spectrum compared to the singleplex network (SRN) and exhibits a noticeable structure that diverges significantly from the simple cross-correlation of stock return network. The significance of the network approach stems from its potential for broad application in the research process involving complex networks. The dimension of multiplex networks forms an integral part of a diverse spectrum of financial market research, thereby expanding the scope of financial market analysis.

Secondly, in the area of financial studies, the financial multiplex network represents the financial market structure in a risk-return relationship with liquidity conditions. It provides a comprehensive view of the financial market by integrating various aspects of interconnections among stocks. The financial multiplex network has richer topological properties than the commonly investigated stock return network. Therefore, the application of the multiplex network approach allows for a more nuanced understanding of the intricacies and complexities of the financial market.

Comprehending the intricate interconnections among assets is of paramount importance for market participants, especially investors. The proposed methodological framework outlines insightful information for decision-makers and investors to improve their portfolio and risk management strategies. The findings of this study are expected to contribute to an understanding of the interconnectivity among stocks, thereby enabling the development of optimal portfolio selection to maximize stock performance under the return-risk ratio, based on the topology and centrality measures of the financial multiplex network. The optimal return-risk trade-offs can be improved through the implementation of an adequate diversification strategy, which can be further optimized through the analysis of the dependency structure of the financial multiplex network.

This paper has been divided into the following parts. Section 2 provides a brief review of the literature. Section 3 describes the data and methodology used in the empirical part, and Section 4 shows the results. The last section discusses the results obtained and provides concluding remarks.

## 2. Literature review

The stock market can naturally be represented as a multiplex network due to the complex and multi-layered interactions among various financial instruments, institutions, and agents. The natural multiplexity of financial markets arises from the coexistence of diverse and

interrelated layers of interaction. This framework allows for the integration of heterogeneous and simultaneously relationships between market participants, capturing their complex interdependencies and the multidimensionality of financial interactions. In the context of financial markets, each layer of the multiplex network can include different types of dependencies [54], highlighting the multifaceted nature of interactions among entities [55]. The interbank market is another example of a naturally multiplexed network representation [13], or, in a broader context, a model of a multilayer financial network that considers different types of interactions among banks, capital markets, and other market participants [56]. Multiplex networks consist of layers that coexist, interact, and evolve within a broader complex system, each characterized by distinct structures and functions [57]. Another study [58] analyzes financial networks as multilayer structures, examining various types of dependencies among financial institutions. The work [23] proposes a framework for a multiplex network, which is based on stock dependencies, sector-specific and location-specific layers.

Research into the financial markets has mainly focused on the configuration of an isolated monoplex network. However, there are studies that take into account the multilayered structure of a multi-relational financial system. The multilayer network approach is a widely used methodology in financial studies, wherein the constituent layers encompass trade, foreign direct investment, and financial market indices [59]; information spillovers, including return spillover, volatility spillover, and extreme risk spillover [60]; Spearman's correlation coefficient, gray relational analysis, and the maximum information coefficient [61]; Pearson, Kendall, tail, and partial correlations [54]; as well as ownership, interlocking directorates, R&D partnerships, and cross-correlation of stock returns [62]. The multilayer network approach has also been applied in the construction of a risk contagion model for financial institutions [18,63–65]. Furthermore, Zhao et al. [66], using the temporal network framework as a special case of a multilayered network, and Lacasa et al. [67] proposed a horizontal visibility graph algorithm to convert a multidimensional time series into the multilayer network.

In real complex systems, there is interdependency between multiple networks [68]. Stock return, volatility, and trading volume are three of the most important features in the context of financial market analysis. In the financial system, the rate of return is derived from the systematic risk premium, which is driven by common factors, and the individual firm-risk premium, which is caused by company-specific factors. A recent analysis by Li et al. [40] reveals that the idiosyncratic return pattern closely resembles the total return pattern during the pre-crisis period, whereas the systematic return trend aligns more closely with the total return patterns during the crash period. The Fama–French model has been used to show, in [41] that the idiosyncratic volatility plays a significant role in explaining the cross-section expected stock returns. Eom and Park [44] investigate the effects of market factors on correlation networks and portfolio diversification. They have found that the MST stock network, based on residuals without a common factor property, plays a key role in building a more diversified portfolio, achieving better performance under the risk–return relationship than the MST network with market factor characteristics. Borghesi et al. [46] point out that applying the firm-specific factor of the stock return network by removing the market mode in clustering leads to less noise, makes the cluster structure more evident, and achieves robustness. Musmeci et al. [47] compare the clustering method for the log-return of the stock price with the market mode and the log-return of the idiosyncratic return. A network based on the idiosyncratic log-returns increases the degree of economic information that the clustering methods retrieve, and means that the clustering is more homogeneous in terms of the number of stocks in the modules. Finally, Todorova [45] investigates the idiosyncratic return through a network approach and finds a positive relationship between network centrality and stock returns.

One of the underlying factors that holds significance in the modeling of financial markets is volatility, which is perceived as a proxy of the riskiness of financial assets. The network of the stock return volatility is one of the most commonly investigated [69,70]. A pioneering study in this domain [71] reveals that the volatility network exhibits more fluctuating degree values than the stock return network. The results of other studies demonstrate a positive correlation between volatility and the largest eigenvalue, and a negative correlation with the number of communities [72], that higher market volatility corresponds to the denser MST-based network [73], that the connectivity of the network is more fragile to selective removal than to random attacks [74], and, finally, that as stocks become increasingly interconnected, their volatility tends to retain a memory of their past behavior [75].

A non-linear causal relationship between the trading volume and the stock return is observed in the financial markets [76–80]. Moreover, Podobnik et al. [81] find evidence for power-law cross-correlations between the absolute values of the log-return of stock price and the logarithmic change in trading volume, concluding that the current price return depends not only on previous returns but also on previous volume change, and vice versa. In the network approach, stock returns and turnover volume have been incorporated into a singleplex network design, using the MST-based multidimensional symbolic method [82]. The trading volume network is the subject of the following studies [83–86].

## 3. Materials and methods

### 3.1. Materials

The data set consists of 465 stocks that were continuously traded on the NYSE or NASDAQ for the period from April 8, 2013 to March 10, 2023. The entire data set was divided into two data sets covering the period starting April 8, 2013 and ending March 8, 2018, with 1240 trading days, for data set 1, and from March 9, 2018 to March 10, 2023, with 1260 trading days (approximately 5 years), for data set 2. The 465 companies remaining in the dataset were included in the S&P500 index on the last day of the time span, and each stock has 2500 data points. The data set for a single observation of each stock contains: i) daily closing price adjusted for splits and dividends; ii) maximum price on the trading day; iii) minimum price on the trading day; and iv) daily trading volume. In addition, data were collected on: a) daily S&P500 index adjusted close prices; and b) interest rates on the 13-week Treasury Bill. All the networks 1) idiosyncratic return; 2) volatility; 3) trading volume; 4) multiplex stock market network, and 5) cross-correlation of stock return were constructed using data set 2. However, the idiosyncratic return network construction methodology requires stock prices, S&P500 indexes, and the 13-week Treasury Bill returns data to be obtained for an additional period encompassing data set 1. These historical data were collected from Yahoo Finance (https://finance.yahoo.com; accessed on 11.03.2023). The five networks mentioned above are binary and undirected. All statistical analyses were performed using the following programs: [87–89]. Additional analyzed data are provided in the S1 File supplementary Information files.

### 3.2. Methods

**3.2.1. Construction of similarity-based networks.** Let $G = (V, E)$ be an undirected and unweighted network consisting of a set of vertices (stocks) $V = \{v_1, v_2, \ldots, v_N\}$, where $V \neq \varnothing$, and a set of edges (relations) between the nodes $E = \{e_1, e_2, \ldots, e_m\}$. The adjacency matrix $A = \{a_{ij}\}$ of a monoplex network is the $N$-dimensional, unweighted, and symmetric matrix with elements

$$a_{ij} = \begin{cases} 1 \text{ if there is an edge between stocks } i \text{ and } j \ (i \neq j) \\ 0 \text{ othewise} \end{cases} \quad (i,j=1,2,\ldots,N) \qquad (1)$$

The adoption of a binary representation of financial networks is justified by several advantages over weighted networks that retain the distance metric on the edges. First, unweighted networks derived from the distance matrix simplify the complexity of the network's structure by reducing the granularity of edge weights. This approach focuses solely on the presence or absence of relationships between assets rather than their precise strength or magnitude. Furthermore, unweighted networks are inherently robust to minor fluctuations and noise in the underlying distance metrics. In financial datasets, the inherent noise and vulnerability to minor fluctuations in correlations can lead to potential overinterpretation of trivial variations in edge weights. By employing binarization, this risk is significantly reduced, thereby enhancing the reliability of the resulting network which effectively delineates only the most critical structural characteristics of the system, thus laying a more dependable foundation for subsequent analysis. Finally, unweighted networks present a uniform framework for comparative analyses across various layers of multiplex network. Weighted networks, by their very nature, complicate such comparisons because of the discrepancies in edge weights that do not have a direct comparability in different contexts. In other words, the added dimension of weighted networks brings with it more complexity, which may hide the basic topology and not allow direct comparison.

**3.2.2. The cross-correlation of stock return network.** Denote by $P_{i,t}$ the adjusted closing price of stock $i$ $(i = 1, 2,\ldots,N)$ at time $t$ $(t = 1, 2,\ldots,T)$. The daily logarithmic returns of the stock prices $r_{i,t}$ can be calculated as

$$r_{i,t} = \ln P_{i,t} - \ln P_{i,t-1} \tag{2}$$

The cross-correlation function based on the two log-return time series for each pair of stocks $i$ and $j$ can be computed using the Pearson correlation coefficient

$$\rho_{ij} = \frac{\langle r_i r_j \rangle - \langle r_i \rangle \langle r_j \rangle}{\sqrt{\left(\langle r_i^2 \rangle - \langle r_i \rangle^2\right)\left(\langle r_j^2 \rangle - \langle r_j \rangle^2\right)}} \tag{3}$$

where the notation $\ldots$ means the average value over a time period $(t = 1,2,\ldots,T)$. The correlation coefficients $\rho_{ij}$ establish a symmetric $N \times N$ matrix $\mathbf{C}$ with $\dfrac{N(N-1)}{2}$ elements and unity on the diagonal. The similarity measure between each pair of stocks requires the three axioms of Euclidean distance to be satisfied: i) positive definiteness; ii) symmetry; and iii) triangular inequality [90]. The correlation coefficients are converted into distance metrics using an appropriate transformation function

$$d_{ij} = \sqrt{2(1 - \rho_{ij})} \tag{4}$$

where $d_{ij}$ denotes the distance metric between companies $i$ and $j$, and ranges from 0, which corresponds to a strong positive correlation coefficient $(\rho_{ij} = 1)$, to 2, which corresponds to a strong negative correlation coefficient $(\rho_{ij} = -1)$.

To filter out the huge amount of information from a fully-connected distance matrix $D = [d_{ij}]$ of dimension $N \times N$, the MST method is applied. The MST is the spanning tree of the shortest length, effectively reducing the information space from $m = \dfrac{N(N-1)}{2}$ edges to $m = N - 1$ most important links, while connecting a set of $N$ vertices without cycles or self-loops. The MST extraction problem is solved using Kruskal's algorithm [91], which spans nodes using a subset of edges with a minimal sum of weights and forms an acyclic graph. The final binary network is obtained through its dichotomization. This network is referred to as the stock return network (SRN).

**3.2.3. MST-based single-layer financial networks.** The following section presents the construction of three monoplex financial networks using three types of relations among assets. *Idiosyncratic return network.* The premium of the firm-specific risk is utilized as a proxy for the idiosyncratic return of the company. To compute the idiosyncratic risk premium, the CAPM model is applied to filter out the common factor of the capital market

$$r_{i,t} = \alpha_i + \beta_i\left(r_{m,t} - r_{f,t}\right) + \varepsilon_{i,t} \tag{5}$$

where $r_{i,t}, r_{m,t}, r_{f,t}$ represent the return of stock $i$ at time $t$, the market return at time $t$, and the risk-free rate at time $t$; $r_{m,t} - r_{f,t}$ refers to the premium of the systematic risk in the capital market; $\alpha_i, \beta_i$ denote the estimated regression coefficient of firm $i$ (the intercept of the security market line) and the exogenous risk of stock $i$, respectively; and $\varepsilon_{i,t}$ corresponds to the residual which represent the idiosyncratic component of the stock return dependent on firm-specific factors. The above are made for the combined time span of data sets 1 and 2. To separate the firm-specific return from the common factor property, the observed returns are regressed using the ordinary least squares (OLS) method. This approach has been applied in the work of numerous researchers in the field: [40,44,45,47,92,93].

Following [92], the daily log return of the S&P500 index is adopted as $r_{m,t}$

$$r_{m,t} = \ln\left(I_t\right) - \ln\left(I_{t-1}\right) \tag{6}$$

where $I_t$ is the daily adjusted closing price of the S&P500 index at time $t$, and the risk-free rate is computed based on the interest rate on the 13-week Treasury Bill expressed in terms of one day

$$r_{f,t} = \frac{TB_t}{365} \tag{7}$$

where $TB_t$ denotes the annual interest rate of the 13-week Treasury Bill.

The residuals $\varepsilon_{i,t}$ without the common factor property are utilized to construct the idiosyncratic return network for data set 2. The logarithmic returns of the residuals are calculated (Eq. (2)) and the rest of the MST-based procedure is followed as for the construction of the cross-correlation of stock return network (Eqs. (3), (4)).

**Volatility network.** To produce the volatility network, the approach adopted in [69,74] is used, in which, for the daily stock price data, the volatility $\sigma_{i,t}$ for each stock $i$ and trading day $t$ is calculated by utilizing the proxy

$$\sigma_{i,t} = 2 \cdot \left|\frac{\max\left\{P_{i,t}\right\} - \min\left\{P_{i,t}\right\}}{\max\left\{P_{i,t}\right\} + \min\left\{P_{i,t}\right\}}\right| \tag{8}$$

where $\max\left\{P_i(t)\right\}$ and $\min\left\{P_i(t)\right\}$ are the highest and lowest price of the trading day.

The selection of the above volatility measure employed in constructing the volatility network is grounded in its ability to accurately capture intraday price movements while remaining computationally efficient. Moreover, the methodology employed is supported by a number of considerations. Firstly, it directly measures the magnitude of price movements within a single trading day, thereby providing a robust proxy for realized volatility. Compared to methodologies based solely on returns or closing prices, this approach incorporates data from the entire trading day, enhancing its sensitivity to transient price dynamics. Secondly, the normalization by the sum of the highest and lowest prices ensures the scale independence of the measure, making it suitable for comparing stocks with varying price levels. This

property is critical for constructing volatility networks, where relationships between stocks are assessed based on relative rather than absolute volatility levels. Finally, the simplicity of the formula makes it computationally efficient and more robust against the noise often present in high-frequency trading data. By using only the highest and lowest prices, the approach avoids potential biases introduced by closing prices, which may not fully exhibit intraday variability due to market microstructure effects or end-of-day trading behaviors.

Then the correlation of the volatility $\sigma_i(t)$ and the distance, applying Eqs. ([3])-([4]), are computed. Next, the Kruskal algorithm is used to build the MST-based volatility network. In the final step, the network is dichotomized.

**Trading volume network.** This type of financial network is based on the cross-correlation of trading volumes between a pair of two stocks. Denote by $v_{i,t}$ the stock trading volume of company $i = 1,2,\ldots,N$ on trading day $t = 1,2,\ \ldots,T$. The construction of this MST-based network is exactly the same as the construction of the stock return correlation network, where the correlation coefficients (Eq. ([3])) are computed for the time series of the logarithmic expression of the trading volume $\{\ln v_{i,t}\}$.

**3.2.4. Multiplex financial network.** The multiplex system of the financial network consists of $N$ vertices and $L$ binary layers $(l = 1,2,\ldots,L)$. In our case there are three layers $(L = 3)$: 1) the idiosyncratic return network; 2) the volatility network, and 3) the trading volume network. The multiplex network can be expressed by the vector of the adjacency matrices of the $L$ layers [94]

$$\mathcal{A} = \left\{ A^{[1]}, A^{[2]}, \ldots, A^{[L]} \right\} \tag{9}$$

$$A^{[l]} = \left\{ a_{ij}^{[l]} \right\} \tag{10}$$

where $a_{ij}^{[l]} = 1$ if nodes $i$ and $j$ are connected by an edge on layer $l$ and $a_{ij}^{[l]} = 0$ otherwise.

In most interconnected complex systems, interactions occur not only among nodes in the same layer, but also between pairs of layers [95]. However, the multiplex network in the financial market consisting of the three network layers that is designed as described in Section 3.2.2, there are no edges between the layers. In a system that displays overlapping edges, the total number of links across all layers is meaningful [25]. Therefore, the multiplex financial network is created by joining the three single network layers. In other words, the multiplex network is obtained from the multilayer structure by combining all links that occur in at least one single layer. Note that possible multiple edges are omitted. The adjacency matrix of the multiplex financial network $\mathcal{A}^{[MP]} = \left\{ a_{ij}^{[MP]} \right\}$ is defined as

$$\mathcal{A}^{[MP]} = \left\{ A^{[l=1]}, A^{[l=2]}, A^{[l=3]} \right\} \tag{11}$$

$$a_{ij}^{[MP]} = \begin{cases} 1 & \text{if } \exists_l a_{ij}^{[l]} = 1 \\ 0 & \text{otherwise} \end{cases} \left( i, j = 1,2,\ldots,N; l = 1,2,3 \right) \tag{12}$$

## 4. Results

### 4.1. Statistical properties of financial networks

[Fig 2] depicts a time-series representation of the residuals ($\varepsilon_{i,t}$) derived from Eq. ([5]), which represent the firm-specific idiosyncratic component of stock returns.

The MST-based graphs of the systematic risk network, volatility network, trading volume network, and multiplex network are shown in Fig 3, and that of the stock return network in Fig 4. Heat maps of the network connections are presented in S1 File Appendix A (Supplementary files). All three layers and the MST-based stock return network demonstrate a typical tree-like structure with a few hub-nodes, while the multiplex network is a graph with a much higher density of connections. A wider assessment of the created networks is carried out on the basis of the global network indicators presented in Table 1.

All networks have the same number of nodes $(N = 465)$. Since all three layers and the stock return network are MST-based networks, they all consist of the same number of edges $(m = N - 1)$, degree (928), and mean degree (1.996). In contrast, the multiplex network with the same number of vertices has more than twice the number of edges and has higher average degrees that exceed 4.5. There is a significantly higher number of maximum degrees $(k_{max} = 35)$ compared to the other networks and a much smaller fraction of pendant nodes (0.03), the so-called leaf fraction, which means the fraction of vertices with no more than one link. For the remaining networks, the leaf fraction is approximately equal to half the number of nodes. This means that the multiplex network does not have any of the typical tree, chain, or star-like network structures, and the most connected vertices in the multiplex network has more connections than the hubs of other networks.

The stock return network and all three layers of the multiplex network are sparse graphs with a density of 0.004. This indicates that the probability of an edge between two randomly

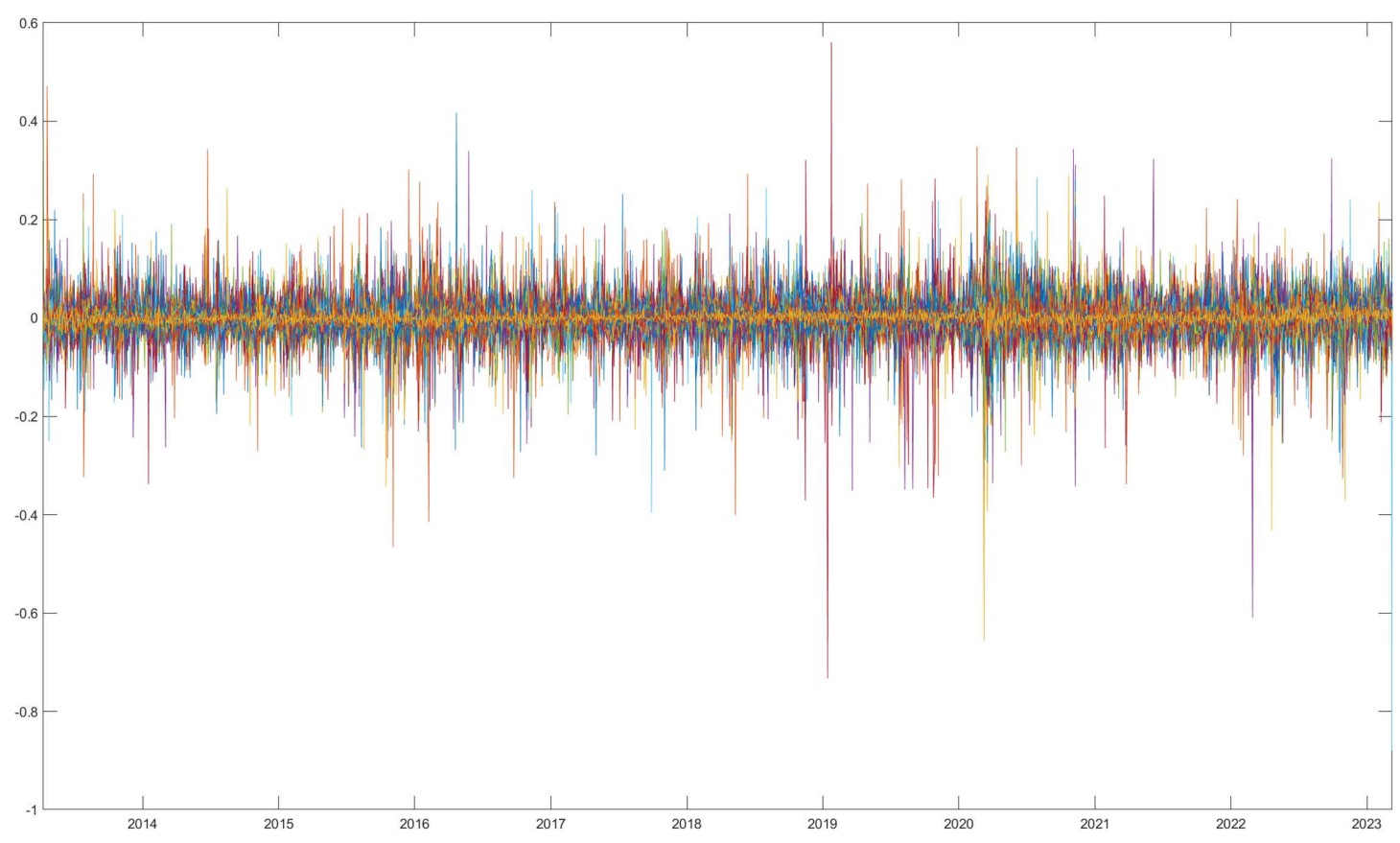

**Fig 2. Time-series of idiosyncratic components of stock returns.**

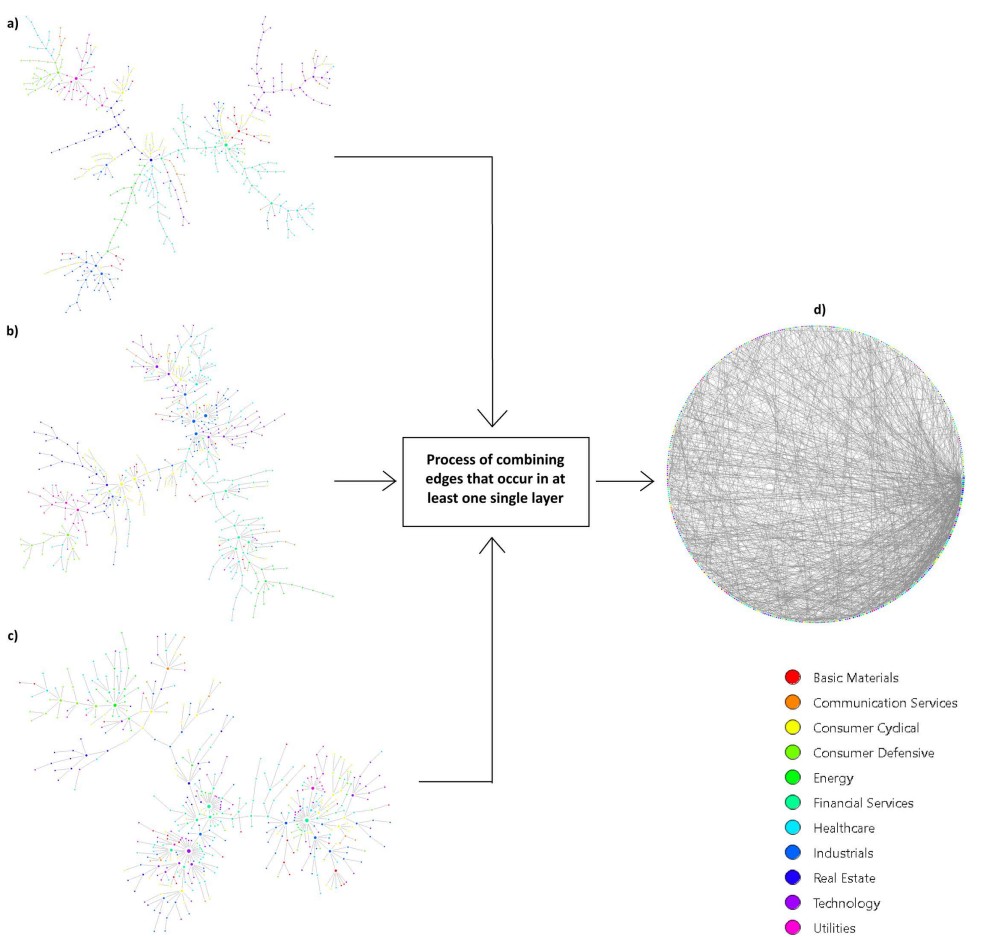

**Fig 3. The MST networks of a) idiosyncratic return; b) volatility; c) trading volume; and d) multiplex network.**

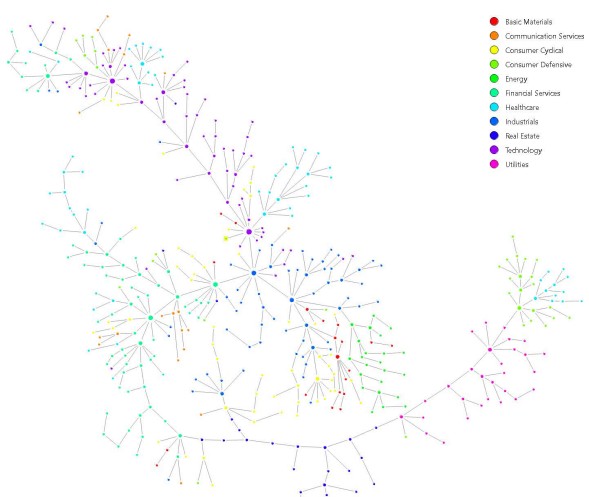

**Fig 4. The MST network of stock returns (monoplex network).**

**Table 1. Network indicators.**

| Indicator | Network | | | | |
|---|---|---|---|---|---|
| | Stock return | Layers | | | Multiplex |
| | | Idiosyncratic return | Volatility | Trading volume | |
| Number of nodes $(N)$ | 465 | 465 | 465 | 465 | 465 |
| Number of edges $(m)$ | 464 | 464 | 464 | 464 | 1 056 |
| Degree $(k)$ | 928 | 928 | 928 | 928 | 2 112 |
| Mean degree $k$ | 1.996 | 1.996 | 1.996 | 1.996 | 4.542 |
| Maximum degree $(k_{max})$ | 13 | 10 | 13 | 26 | 35 |
| Fraction of pendant nodes $(FPN)$ (number of nodes with degree $k=1$ in parentheses) | 0.538 (250) | 0.454 (211) | 0.574 (267) | 0.626 (291) | 0.028 (13) |
| Network density $(\rho)$ | 0.004 | 0.004 | 0.004 | 0.004 | 0.010 |
| Mean geodesic distance $(L)$ | 15.048 | 19.698 | 12.703 | 10.384 | 4.201 |
| Network diameter $(s)$ | 41 | 46 | 33 | 25 | 9 |
| Graph radius $(r)$ | 21 | 23 | 17 | 13 | 6 |
| Clustering coefficient $(C)$ | 0.000 | 0.000 | 0.000 | 0.000 | 0.152 |
| Transitivity $(T)$ | 0.000 | 0.000 | 0.000 | 0.000 | 0.076 |
| Network tree length $(NTL)$ | 0.727 | 0.939 | 0.708 | 0.940 | – |
| Mean occupation layer $(MOL)$ | 9.622 | 13.854 | 8.559 | 7.888 | 2.834 |
| Network degree centralization index $(NDCI)$ | 2.382% | 1.733% | 2.382% | 5.196% | 6.593% |
| Network closeness centralization index $(NCCI)$ | 7.552% | 5.855% | 9.559% | 11.712% | 22.186% |
| Network betweenness centralization index $(NBCI)$ | 62.474% | 63.314% | 63.749% | 70.265% | 18.493% |
| Modularity $(Q)$ | 0.904 | 0.902 | 0.901 | 0.902 | 0.555 |
| Number of communities $(NC)$ | 19 | 19 | 20 | 20 | 14 |
| Exponent of power-law $(\gamma)$ ( $p$ -value in parentheses) | 3.023 ( $p$ =0.520) | 4.921 ( $p$ =0.691) | 2.534 ( $p$ =0.836) | 2.781 ( $p$ =0.117) | 3.029 ( $p$ =0.714) |
| Hub node tickers, based on 99th percentile (degree in the parentheses) | APH (13); MSFT (12); AME (11); AMP (11); PRU (10) | PRU (10); CMS (9); SFG (8); ITW (7); PEP (7) | AME (13); ETN (13); HON (12); ADBE (11); DRI (11) | AMP (26); MSFT (26); PRU (21); XOM (14); WEC (12) | AMP (35); MSFT (30); PRU (29); ETN (21); AME (19) |

selected vertices is 0.4%. By contrast, the multiplex network is more dense and the probability is 1%.

The tree-like structure of the stock return network and the three layers is confirmed by the high value of the mean geodesic distance, which exceeds the value of 10. This means that between two selected nodes in these networks there are, on average, 9 intermediate nodes for

the trading volume network and approximately 18–19 for the idiosyncratic risk network. The average shortest path length for the multiplex network is only 4.2 (approximately 3 intermediate nodes between two stocks), which indicates that the multiplex network is more compact than the other graphs. The compactness of the multiplex network is confirmed by the low values of the network diameter (9) and graph radius (6) compared to the other networks, for which the values range from 25 to 46 and from 13 to 23, respectively. It is worth noting that the diameter of the multiplex network (9) is lower than the minimal value of the graph radius of the remaining networks (13). This indicates that the FMN presents short cuts.

Since all the networks except the multiplex one are MST-based networks, which are graphs without loops, the global clustering coefficient and transitivity are equal to 0 (there are no triangles). By contrast, we observe positive values of the above indicators for the FMN. The likelihood of two adjacent neighbors of a vertex being clustered together is 15.2%. It should be emphasized that, based on the density, mean geodesic distance, network diameter and clustering coefficient, the properties of a small-world network may be exhibited by the multiplex network. The verification of this issue is the subject of Section 4.2.

The net tree length (NTL) is used to assess the length of the MST-based networks. For this reason, the multiplex network is not considered, because it does not meet the minimum spanning tree criteria, due to $m \neq N - 1$ and the graph containing loops. We observe that the NTL is substantively shorter for the SRN and for one layer – the volatility network $\left( NTL \approx 0.7 \right)$. As the NTL increases, the MST network structure becomes less tightly held. Comparing the shrunken trees of stock returns and volatility with those of idiosyncratic return and trading volume suggests that stock prices and volatility tend to move in the same direction more strongly than the firm-specific factor of the stock return and the turnover of assets. The SRN shows a higher level of correlation than the idiosyncratic return network layer because the construction of this layer is based on the common factor-free stock return.

The FMN has a significantly lower value for the mean occupation layer (MOL) compared to all three layers of the multiplex network and the SRN. This suggests that the FMN is more densely connected than the other networks, with each vertex located only approximately two nodes away from the center. A lower MOL value for the multiplex network indicates that the transmission of all information provided by each of the three layers from the central vertex to the other stocks requires fewer intermediate nodes than is required for the other networks. This is a consequence of the higher density of the multiplex network, in which local connectivity of the network as connected triples of nodes is allowed. Nonetheless, MOL values ranging from 7.9 to 13.9 for the SRN and all three layers confirm that these MST-based networks exhibit a tree-like structure.

Based on the network degree centrality index (NCDI), it can be concluded that the FMN network is the most centralized $\left( NDCI = 6.6\% \right)$ of all the networks considered, and that the idiosyncratic return network is the least centralized $\left( NDCI = 1.7\% \right)$. In other words, the most connected nodes in the FMN network have a higher degree than in the other networks. This confirms the finding that the FMN network has a significantly higher maximum degree $\left( k_{max} \right)$ than the other networks.

The compactness of the FMN is confirmed by the network closeness centrality index (NCCI), which is approximately 2–4 times larger than the NCCI of the remaining networks. This implies that there are several vertices in the FMN that are very close to other nodes, the so-called short cuts. On the other hand, the FMN shows a much lower level of betweenness than the other networks. The network betweenness centralization index (NCBI) is 18.5%, while it ranges from 62.5% to 70.3% for the other networks. The observed higher centralization of the SRN and all three layers of the multiplex network in terms of betweenness is due to their tree-like structures, where several vertices act as central intermediate nodes. In contrast,

the FMN network exhibits a much more evenly distributed betweenness centrality. It should be stressed that the NCCI is larger than the NCBI only for the multiplex network, which clearly indicates a different shape and structure of this network.

Modularity (Q) values for all four networks – SRN, idiosyncratic return, volatility, and trading volume – range from 0.901 to 0.904, indicating that the networks have a strong community structure. The modularity result obtained is significantly high. However, the multiplex network has a lower modularity value of 0.555, but still shows significant community partitioning $(Q > 0.3)$. There is also a difference in the number of communities. The Louvain Fast Unfolding algorithm [96] identifies fewer communities (14) for the multiplex network than for the other networks (from 19 to 20) for the same number of nodes ( $N = 465$ ). This is the result of the clustering tendency observed in the multilayer network but not in the other networks.

To assess whether the analyzed networks are scale-free, 930,000 iterations (2,000 iterations per vertex) a powerful statistical approach developed in [97] are performed. The power-law exponents of all the networks range from 2.534 (volatility layer) to 4.921 (idiosyncratic return layer) and the corresponding $p$ -values are larger than 0.1, indicating that both networks, the SRN and the FMN, and each layer of the multiplex network obey a power-law vertex degree distribution. This implies that these networks have a scale-free structure, which means that they are composed of self-similar structures at different scales. The power-law degree distribution indicates that the network is highly heterogeneous, meaning that there are a few highly connected hub-nodes that play a crucial role in the overall connectivity of the network, and many vertices with low degree. This observed nature, which is typical of a scale-free network, distinguishes the analyzed financial networks from random networks in the sense of E–R random graphs, whose degree distribution follows the Poisson distribution. S1 File Appendix B (Supplementary files) provides more information on the analysis of the power-law degree distributions, such as the KS test statistics, and the lower bound ( $x_{min}$ ), and a graphical presentation of the degree distributions using the complementary cumulative distribution function (CDF). It should be noted that a power-law degree distribution is also observed for the overlapping degree of the multiplex network

$$o_i = \sum_{l=1}^{L} k_i^{[l]} \tag{13}$$

where the exponent $(\gamma)$ is 2.972 ( $p$ -value = 0.144).

Since the constructed financial multiplex network is a fully multiplexed system in which every node is presented in each of the three layers, a suitable measure of the distribution of the degree of the vertex $i$ among the separate layers is the multiplex participation coefficient (MPC) [54,94]

$$P = \frac{1}{N} \sum_{i=1}^{N} P_i \tag{14}$$

$$P_i = \frac{L}{L-1} \left[ 1 - \sum_{l=1}^{L} \left( \frac{k_i^{[l]}}{o_i} \right)^2 \right] \tag{15}$$

The MPC measures the distribution of the edges connected to node $i$ across multiple layers. $P_i$ takes values in the range $0,1$, where 1 indicates that node $i$ has the same number of edges on each of the $L$ layers, and 0 means that vertex $i$ has connections in only one layer. The average multiplex participation coefficient (MPC) is equal to $P = 0.933$, indicating that the

participation of most vertices by degree is uniformly distributed among the three layers of the multiplex network. The MPC distribution is depicted in Fig 5.

Table 2 reports the degree–degree correlation analysis with the Quadratic Assignment Procedure (QAP) results. The results show substantial negative assortativity coefficients for all networks. This means that the indicated networks display disassortative behavior, where stocks tend to be adjacent to assets with dissimilar degrees. This can be interpreted as a tendency towards heterogeneity in the connections between vertices. Although the multiplex network has a negative assortativity coefficient, it is closest to 0 with a value of -0.072, indicating that there is a poor tendency to connect vertices with other nodes of dissimilar degrees.

It should be emphasized that the QAP analysis [98], performed with 930,000 iterations, demonstrates a high probability (equal to or close to 1) that the observed assortativity coefficient values are significantly different from the expected value close to 0, which characterizes a non-randomly created network.

A preliminary analysis of the networks confirms the presence of sectoral assortativity (see Table 3). A positive assortativity coefficient indicates that networks exhibit assortative behavior, characterized by a tendency to form connections between assets belonging to the same

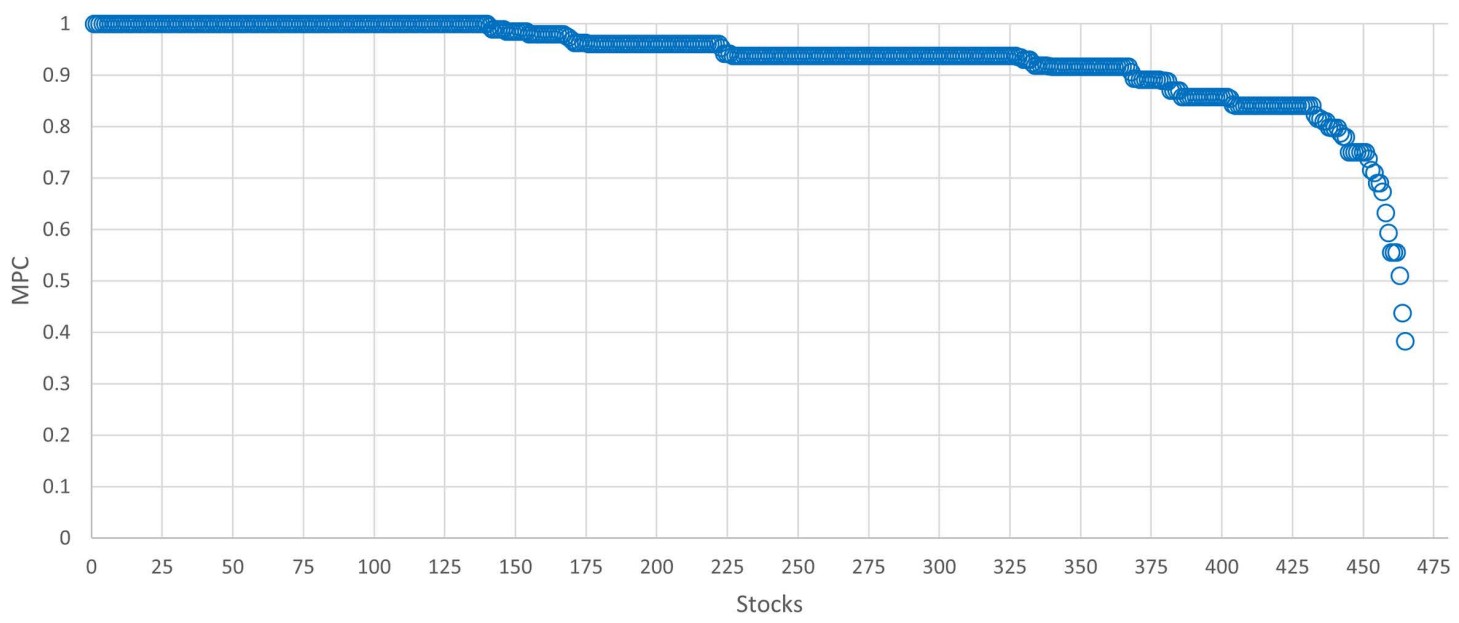

**Fig 5. Distribution of the multiplex participation coefficient.**

**Table 2. Results of degree assortativity mixing.**

| Network | Observed r | Expected r | Std. Dev. | P (expected r ≥ observed r) | P (expected r = observed r) | P (expected r ≤ observed r) |
|---|---|---|---|---|---|---|
| Multiplex | -0.072 | -0.003 | 0.030 | 0.998 | 0.000 | 0.003 |
| Idiosyncratic return | -0.161 | -0.003 | 0.046 | 1.000 | 0.000 | 0.000 |
| Volatility | -0.207 | -0.004 | 0.045 | 1.000 | 0.000 | 0.000 |
| Trading volume | -0.180 | -0.004 | 0.043 | 1.000 | 0.000 | 0.000 |
| Stock return | -0.170 | -0.004 | 0.045 | 1.000 | 0.000 | 0.000 |

Number of QAP iterations = 930,000.

**Table 3. Results of sector assortativity mixing.**

| Network | Observed r | Expected r | Std. Dev. | P (expected r $\geq$ observed r) | P (expected r = observed r) | P (expected r $\leq$ observed r) |
|---|---|---|---|---|---|---|
| **Multiplex** | **0.539** | -0.003 | 0.030 | 0.000 | 0.000 | 1.000 |
| **Idiosyncratic return** | **0.811** | -0.003 | 0.046 | 0.000 | 0.000 | 1.000 |
| **Volatility** | **0.618** | -0.004 | 0.046 | 0.000 | 0.000 | 1.000 |
| **Trading volume** | **0.425** | -0.005 | 0.046 | 0.000 | 0.000 | 1.000 |
| **Stock return** | **0.755** | -0.004 | 0.046 | 0.000 | 0.000 | 1.000 |

Number of QAP iterations = 930,000.

sector. This property is most prominent in the idiosyncratic return layer. Previous research has demonstrated that stocks in correlation-based networks consistently form clusters closely aligned with economic sector classifications [6,28,46,72,90,92,99–103]. Likewise, the results are statistically significant when employing the QAP approach.

## 4.2. The small-world structure of the multiplex network

The small-world characteristics of a corporate network [104] are widely acknowledged as a stylized fact [105] and refers to the idea that any two nodes in a large complex network can be linked to each other through a relatively short chain of intermediaries. The multiplex network is the only one considered here because the other networks do not exhibit the property of cliquishness. In other words, the global clustering coefficient and transitivity are only greater than zero for the multiplex network (see Table 2), which means that the other networks do not show the small-world phenomenon. The indicators utilized to evaluate the small-world character of the network are presented in Table 4.

The multiplex network is numerically large and sparsely connected, since $N \gg k_{max} \gg 1$, where $N = 465$ and $k_{max} = 35$. We compare the values of $C_{actual}$ and $L_{actual}$ with the asymptotic approximation of the clustering coefficient and the mean shortest path length for the equivalent E–R random network with the same number of nodes and edges. Since the relations $C_{actual} \gg C_{random}$ and $L_{actual} \sim L_{random}$ are satisfied, indicating that the multiplex network

**Table 4. Small-world quantities for the multiplex network.**

| Indicator | value |
|---|---|
| $\rho$ | 0.010 |
| $s$ | 9 |
| $L_{actual}$ | 4.201 |
| $L_{random}$ | 4.059 |
| $C_{actual}$ | 0.152 |
| $C_{random}$ | 0.010 |
| $C_{actual} / \rho$ | 15.2 |
| $\gamma^{ws}$ | 15.561 |
| $\lambda$ | 1.035 |
| $S^{ws} = \gamma^{ws} / \lambda$ | 15.034 |

$\rho$ – Network density, $s$ – Network diameter, $L_{actual}$ - Mean geodesic distance (average shortest path length), $C_{actual}$ – Clustering coefficient.

exhibits the small-world property. Additional evidence of the small-world phenomenon leads to a similar conclusion. Since $\lambda \sim 1$ and $\gamma^{WS} \gg 1$ are observed, the ratio of $\gamma^{WS}$ to $\lambda$ is much larger than one ( $S^{WS} = 15.034$ ). S1 File Fig. C1 presented in Appendix C (Supplementary files) depicts the relationship between $\gamma^{WS}$ and $\lambda$.

The small-world property of the multiplex network means that there are short cuts that reduce the distance between nodes that are not directly connected. These long-range connections allow for closer relationships between stocks and shorter paths between vertices that are otherwise separated by many intermediate nodes. Short cuts play a crucial role in creating a more compact and interconnected network. The presence of short cuts in a multiplex network results from the compactness created by several nodes that have a high degree of connectivity and act as hubs within the graph. Although the existence of hub-nodes has been detected for all the networks (the power-law property), only the FMN is a scale-free and small-world network.

## 4.3. Similarity between layers and networks

Similarity refers to the degree of association between two or more networks, which is measure of how the networks are related. The similarity coefficient can be used to evaluate the similarity between two different networks by comparing the sets of edges that they have in common. The Pearson correlation coefficient, the Jaccard index and the Czekanowski–Sørensen–Dice similarity coefficient, are utilized.

Table 5 reports the correlation coefficients between the individual networks. The correlation between each layer and the multiplex network is moderate $(\rho = 0.661)$, which suggests that the different layers contribute distinct information towards the final form of the multiplex network. In comparison, the correlation between the FMN and the SRN is lower, at 0.514. This is due to the varied relationship between the different layers of the multiplex network and the stock return network, where the layer of the network based on the firm-specific factor of the stock return is the most similar to the SRN, while the layer of the graph based on the trading volume is the least similar. Moreover, the similarity between the three layers is moderate ( $\rho$ is approximately 0.4) or weak (below 0.3). It should be pointed out that, based on the QAP analysis, all the correlation coefficients are statistically significant at the level of 0.1%. A better measure of similarity between networks is the matching measure. Table 6 presents the Jaccard similarity measure (lower triangle) and the Czekanowski–Sørensen–Dice similarity coefficient (upper triangle).

The Jaccard similarity between each layer and the FMN is 0.439, which means that 43.9% of the edges overlap, while the remaining 56.1% of the links are derived from the other two layers. The edge overlapping between the FMN and the SRN is less than one third (362/1,158 = 0.313). As we saw in the analysis based on the correlation coefficient, the smallest similarity

**Table 5. Pearson correlation coefficients.**

| Network | Multiplex | Idiosyncratic return | Volatility | Trading volume |
|---|---|---|---|---|
| **Multiplex** | | | | |
| **Idiosyncratic return** | 0.661[*] | | | |
| **Volatility** | 0.661[*] | 0.426[*] | | |
| **Trading volume** | 0.661[*] | 0.229[*] | 0.240[*] | |
| **Stock Return** | 0.514[*] | 0.699[*] | 0.463[*] | 0.225[*] |

[*]denotes significant level of 0.1% based on the QAP with 9,300 iterations, where the probability that the observed correlation coefficient is greater than expected is 1.000.

**Table 6. Jaccard and Czekanowski–Sørensen–Dice matching coefficients.**

| Network | Multiplex | Idiosyncratic return | Volatility | Trading volume | Stock Return |
|---|---|---|---|---|---|
| **Multiplex** | | 0.611[*] | 0.611[*] | 0.611[*] | 0.476[*] |
| **Idiosyncratic return** | 0.439[*] | | 0.429[*] | 0.233[*] | 0.700[*] |
| **Volatility** | 0.439[*] | 0.273[*] | | 0.244[*] | 0.466[*] |
| **Trading volume** | 0.439[*] | 0.132[*] | 0.139[*] | | 0.228[*] |
| **Stock Return** | 0.313[*] | 0.539[*] | 0.303[*] | 0.129[*] | |

Upper triangle: Czekanowski matching coefficients; Lower triangle: Jaccard matching coefficients.

[*]denotes significant level of 0.1% based on the QAP with 9,300 iterations, where the probability that the observed correlation coefficient is greater than expected is 1.000.

to the SRN is shown by the layer of the trading volume network (12.9%), and the greatest similarity is shown by the layer of the multiplex network based on the firm-specific factor of stock return (53.9%). When comparing the separate layers of the multiplex network, the edge overlap is relatively low, indicating their mutual diversity. The values of the Czekanowski–Sørensen–Dice coefficients confirm the above observations. The Sørensen–Dice coefficient tends to give slightly higher similarity values than the Jaccard similarity index because the Sørensen–Dice measure gives more weight to the common elements and less weight to the non-common elements. After conducting 9,300 QAP iterations, it was determined that all the similarity measures in Table 6 exhibit statistical significance. Specifically, the observed matching coefficients are larger than would be expected by random chance with a probability of 1.0. It is noteworthy that the similarity measures, including Pearson correlation coefficients, the Jaccard index, and Czekanowski-Sørensen-Dice matching coefficients, for each layer with the FMN, yield identical values. This phenomenon results from aggregating three layers with varying similarities into a multiplex form, where some edges overlap across layers, and each layer contains the same number of edges $(N-1=464)$ due to the MST properties.

Computing the Jaccard similarity coefficient for the four networks is recommended, considering all three layers and the multiplex network. The formula in the following can be used to extend the Jaccard coefficient to more than two sets

$$J_{(G_1, G_2, \ldots, G_s)} = \frac{\left| E_{G_1} \wedge E_{G_2} \wedge \ldots \wedge E_{G_s} \right|}{\left| E_{G_1} \vee E_{G_2} \vee \ldots \vee E_{G_s} \right|} \tag{16}$$

where $E_{G_1}, E_{G_2}, \ldots, E_{G_s}$ represent the edges of each network, and $s$ is the number of sets. Table 7 shows the Jaccard index for all three layers.

The Jaccard index for all three layers and the multiplex network is 7.95%, which means that, of the 1,056 edges contained in all four networks together, only 84 overlap (84/1,056 = 0.0795). Fig 6 depicts a graphical illustration of the edge overlapping for the multiplex network and its three layers, as well as for the FMN with the SRN. The Jaccard index for three layers and for each pair of layers ranges from 10.2% to 11.5%. This indicates a higher proportion of overlapping edges within each pair of layers compared to the multiplexed representation.

**Table 7. Jaccard index for two or more sets.**

| Networks/ layers | Multiplex | Idiosyncratic return & Volatility | Idiosyncratic return & Trading volume | Volatility & Trading volume |
|---|---|---|---|---|
| **Three layers** | 0.0795 | 0.1152 | 0.1024 | 0.1031 |

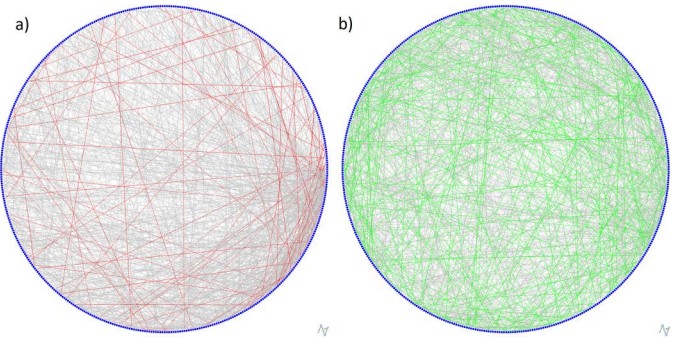

**Fig 6. Edge overlapping for a) FMN with all three layers (red edges; 7.95%); b) FMN with the SRN (green edges; 31.3%).**

Another approach is the edge overlapping ratio $\left(EOR\right)$ in the compared networks, which is defined as the ratio of the common edges found in $s$ consecutive networks and the maximum number of edges in the network

$$EOR(s) = \frac{\left| E_{G_1} \cap E_{G_2} \cap \ldots \cap E_{G_s} \right|}{\max\left\{ m_{G_1}, m_{G_2}, \ldots, m_{G_s} \right\}} \tag{17}$$

where $E_{G_a}, E_{G_b}, \ldots, E_{G_s}$ represent the sets of links of networks $G_a, G_b, \ldots, G_s$; $m_{G_1}, m_{G_2}, \ldots, m_{G_s}$ are the number of edges in each network. The EOR index is an original proposal defined in this study.

It should be noted that the edge overlapping ratio (EOR) proposed in this study for the FMN and its three layers corresponds exactly to the value of the Jaccard index (EOR = 7.95%). This convergence is because the denominator of the EOR – the maximum number of edges of one network – corresponds to the denominator of the Jaccard index, i.e., the union of the sets of edges of all the networks considered $\left(m = 1056\right)$. On the other hand, the numerators of both indexes are the same by definition.

However, the EOR for the FMN with the SRN is different from Jaccard's ratio and is 34.4%. In other words, the overlapping edges cover 34.3% of the maximum number of links in one of these networks (362/1056). The edge overlapping ratio for comparisons between individual layers of the multiplex network and the SRN is included in S1 File Appendix D (Supplementary files).

Another quantity used to analyze multiplex networks is the correlation of degrees across layers. In order to determine the inter-layer degree correlations, the mutual information of the degree sequences can be used. The inter-layer mutual information between the degree distributions of two layers is defined as follows [67]

$$I = \sum_{k^{[\alpha]}} \sum_{k^{[\beta]}} P\left(k^{[\alpha]}, k^{[\beta]}\right) \log \frac{P\left(k^{[\alpha]}, k^{[\beta]}\right)}{P\left(k^{[\alpha]}\right) P\left(k^{[\beta]}\right)} \tag{18}$$

where $P\left(k^{[\alpha]}, k^{[\beta]}\right)$ is the joint probability that a node has degree $k$ in both layers

$$P\left(k^{[\alpha]}, k^{[\beta]}\right) = \frac{N_{k^{[\alpha]}, k^{[\beta]}}}{N} \tag{19}$$

where $N_{k^{[\alpha]},k^{[\beta]}}$ denotes the number of vertices with degrees $k^{[\alpha]}$ and $k^{[\beta]}$ in layers $\alpha$ and $\beta$, respectively. The higher the value of the inter-layer mutual information, the more correlated are the degree distributions of the two layers.

The inter-layer mutual information $I$ for individual pairs of the three layers is as follows: 0.419 for the idiosyncratic return–volatility pair; 0.449 for volatility–trading volume; and 0.359 for idiosyncratic return–trading volume. The average quantity among all possible pairs is 0.409, which means that the degree distribution is not strongly correlated between the multiplex layers.

## 4.4. Multiplex network regression analysis

Another method of testing the importance of different layers is to perform the Multiple Regression Quadratic Assignment Procedure (MR-QAP) network regressions [106]. In this study, three multiple regression models are proposed to determine whether the structure of each layer explains i) the financial multiplex network and ii) the stock return network

$$A^{[FMN]} = \alpha + \beta_1 A^{[1]} + \beta_2 A^{[2]} + \beta_3 A^{[3]} \tag{20}$$

$$A^{[SRN]} = \alpha + \beta_1 A^{[1]} + \beta_2 A^{[2]} + \beta_3 A^{[3]} \tag{21}$$

$$A^{[FMN]} = \alpha + \beta_1 A^{[SRN]} \tag{22}$$

where $A^{[FMN]}$ is the adjacency matrix of the financial multiplex network; $A^{[SRN]}$ the adjacency matrix of the stock return network; $\alpha$ indicates a constant; $\beta_1$, $\beta_2$, $\beta_3$ mean regression coefficients; and $A^{[1]}, A^{[2]}, A^{[3]}$ denote the adjacency matrices of the idiosyncratic return network, the volatility network, and the trading volume network, respectively.

These regression models estimate how the layers of the multiplex network affect the two networks – the multiplex financial network and the stock return network. The objective of Model 1, defined in Eq. (20), is to assess the effect of its individual components (layers) on the generation of the FMN. Model 1 serves as the initial framework for Model 2, which is described by Eq. (21). Both Model 1 and Model 2 are used to evaluate the differences in the strength of the impact that each layer of the multiplex network has on the FMN and SRN, respectively. To evaluate the statistical significance of the specified models, the MR-QAP test is used, which, as an extension of the bivariate QAP model, is a permutation test for multiple linear regression model coefficients for network data organized in square matrices [107]. The Double-Semi-Partialing (DSP) approach introduced by Dekker et al. [107], which is a residual permutation method, is utilized. Under permutations, the DSP method reduces the potential effects of multicollinearity between the focal variable and the control variables.

Table 8 reports the results of the estimation of the three cross-network regression models formulated by Eqs. (20)-(22). Based on the regression coefficients of Model 1, the positive effect of all three layers on the multiplex network can be observed, with the trading volume network layer having the strongest effect on the FMN. Positive regression coefficients are reported for Model 2. The strongest effect on the SRN is the multiplex network layer concerning idiosyncratic return. The explanation for this is that the rate of return is the sum of the systematic and the idiosyncratic risk premiums. It is important to emphasize that the proportion of variance in the dependent variable that can be explained by the independent variables in Model 2 is $R^2 = 52.4\%$. This means that about half of the information contained in the SRN

**Table 8. Network regression results.**

| MODEL 1 | Dependent variable: Multiplex network | Coefficient | P (expected> = observed) | P (expected = observed) | P (expected < = observed) |
|---|---|---|---|---|---|
| | Constant | 0.002*** (0.000) | – | – | – |
| | Idiosyncratic return | 0.585*** (0.002) | 0.000 | 0.000 | 1.000 |
| | Volatility | 0.571*** (0.002) | 0.000 | 0.000 | 1.000 |
| | Trading volume | 0.723*** (0.001) | 0.000 | 0.000 | 1.000 |
| | *F-test* | 340,829.926*** | 0.000 | 0.000 | 1.000 |
| | $R^2$ | 0.826 | | | |
| MODEL 2 | Dependent variable: Stock return network | Coefficient | P (expected> = observed) | P (expected = observed) | P (expected < = observed) |
| | Constant | 0.001*** (0.000) | – | – | – |
| | Idiosyncratic return | 0.607 *** (0.002) | 0.000 | 0.000 | 1.000 |
| | Volatility | 0.195*** (0.002) | 0.000 | 0.000 | 1.000 |
| | Trading volume | 0.039*** (0.002) | 0.000 | 0.000 | 1.000 |
| | *F-test* | 70,014.112*** | 0.000 | 0.000 | 1.000 |
| | $R^2$ | 0.524 | | | |
| MODEL 3 | Dependent variable: Multiplex network | Coefficient | P (expected> = observed) | P (expected = observed) | P (expected < = observed) |
| | Constant | 0.006*** (0.000) | – | – | – |
| | Stock return network | 0.774*** (0.003) | 0.000 | 0.000 | 1.000 |
| | *F-test* | 77,587.509*** | 0.000 | 0.000 | 1.000 |
| | $R^2$ | 0.264 | | | |

Number of observations $N(N-1) = 215,760$; standard error for each coefficient in parentheses; MR QAP – Double-Semi-Partialing (DSP) method; number of iterations for each model: 46,500;

***Denotes significance at the 0.1% level (based on standard regression analysis).

is determined by the transfer of information embedded in the three layers of the financial multiplex network. The remaining variability is explained by other factors not included in Model 2. It should be pointed out that, in contrast to Model 1, where the trading volume layer has the strongest influence on the formation of the FMN, this layer in Model 2 exerts the weakest influence on the SRN. This change underscores the differences between the FMN and SRN.

The third model (Model 3) deals with the impact of the SRN on the FMN. The modeled reverse relationship exhibits a much lower overall measure of goodness-of-fit, expressed by the coefficient of determination $(R^2 = 26.4\%)$, than Model 2. However, Models 2 and 3 are not directly comparable, as one contains a multiplex network and the other the three layers of the multiplex network.

In all the regression models, the F-test value and the regression coefficients for all variables are statistically significant, while, by applying 46,500 iterations of Double-Semi-Partialing of MR-QAP, the observed quantities are statistically significantly larger than expected under the random chance assumption.

## 4.5. Robustness to network failure

Since all networks – the FMN, its layers, and the SRN – are scale-free graphs, we can determine the network's robustness to random vertex failures. Evaluation in this regard has been carried out using the critical threshold of network failure, which, by applying the Molloy-Reed criterion to form a giant component in a network, can be defined as follows [108]

$$f_c = 1 - \frac{1}{\frac{\langle k \rangle^2}{\langle k \rangle} - 1}$$

(23)

where $\langle k \rangle^2$ and $\langle k \rangle$ denote the second and first moments of the degree distribution, respectively.

This critical threshold $f_c$ indicates a finite fraction of random node removal to collapse the largest component structure. In other words, the random removal of a $f_c$ fraction of vertices will result in the fragmentation of the network.

Table 9 reveals that the FMN exhibits the most robust structure against random failures. In order for the financial multiplex network to fall apart, one would have to remove 84% of its vertices, while to damage the giant component of the stock return network, it is enough to randomly remove 58.6% of the nodes. It should be emphasized that the result robustness of the SRN is consistent with those obtained by [37], where for the MST-based cross-correlation network the critical robustness coefficient is $f_c = 0.61$ and $f_c = 0.56$, respectively, depending on the period. This suggests that the financial market demonstrates resilience to failures in the aftermath of random stock removal.

## 4.6. Robustness analysis

The fundamental analysis was conducted through the simplification of a network approach, which entailed the dichotomization of the MSTs obtained using the distance metric $d_{ij}$ and the subsequent construction of an unweighted multiplex network. The robustness test is applied to the relaxation of the network binarization assumption, resulting in the formation of a multiplex network, its corresponding individual layers, and the SRN as a weighted network. Fig 7 illustrates the visualization of the multiplex graph (FMN) using a force-directed layout, which was generated with the ForceAtlas2 algorithm (node colors correspond to sector affiliation). In the presented graph, four companies (PRU, MSFT, AMP, ETN) have been

**Table 9. Critical threshold for robustness to failure.**

| Network | SRN | FMN | Layer 1 | Layer 2 | Layer 3 |
|---|---|---|---|---|---|
| $f_c$ | 0.586 | 0.840 | 0.458 | 0.637 | 0.751 |

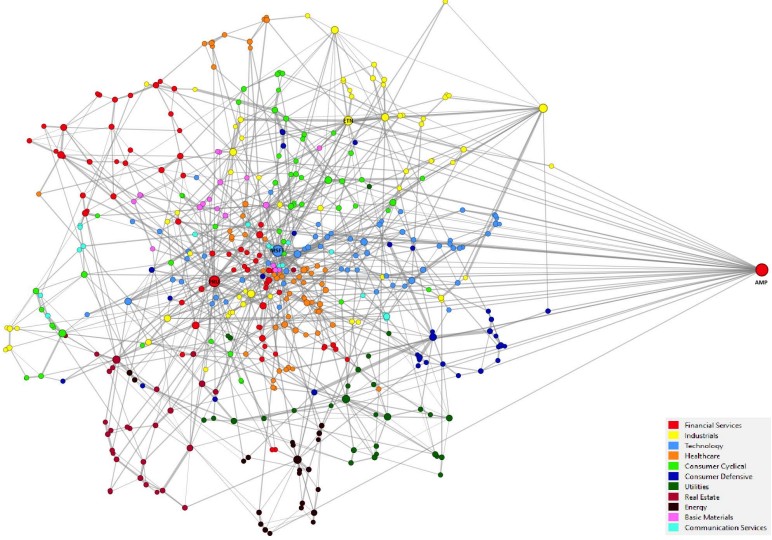

**Fig 7. Weighted multiplex network.**

highlighted, as their degree centrality in the weighted multiplex network exceeds the threshold value of 0.05 (degree centrality > 0.05). Two of these firms belong to the Financial Services sector, while one represents the Technology sector and another the Industrials sector. Notably, the AMP node stands out due to both its high degree of connectivity and its considerable distance from other vertices.

It should be stressed that comparisons of the statistical properties of the networks were omitted, as weighted and unweighted networks inherently differ. The robustness test is applied to the analyses presented in Sections 4.3, 4.4, and 4.5.

Table 10 shows the results of the correlations between the weighted networks. The results do not reveal significant differences compared to the basic analyses. It should be pointed out that the assessment of network similarity, as measured by Jaccard similarity, Czekanowski-Sørensen-Dice matching coefficients, and EOR, yields identical values. This is because the evaluation considers only the presence of edges between nodes in the network.

The results obtained for network regression (Table 11) and the threshold for robustness to failure (Table 12) in weighted networks are similar to those obtained for unweighted networks. This indicates the robustness of the results with respect to the assumption of dichotomization of the networks at the final stage of their construction.

## 4.7. Out-of-sample assessment of the dependency structure representation

In order to assess which financial network has a structure that better represents stock price movements, a regression analysis has been performed employing the ordinary least squares method. The dependent variable is the Sharpe ratio $\frac{r_i}{\sigma_i}$, as proxy for stock performance [34], using out-of-sample data of the stock returns over the 252 days, with a one-year lag from the last data point used to construct the financial networks. The time frame for the out-of-sample data is March 13, 2023, to March 12, 2024. The explanatory variables employed are two centrality measures calculated for the two networks, respectively FMN and SRN (for the in-sample data): 1) degree centrality and 2) eigenvector centrality. Degree centrality is a simple measure that captures connections in the nearest vertex neighborhood in the network. In contrast, eigenvector centrality is a more complex measure that considers the wider context of the structure of the connections in the network. A similar approach was applied by [101], investigating the correlation between the in-sample centrality and the out-of-sample Sharpe ratio. Due to changes in listed shares over the out-of-sample period, the number of observations is 456. Table 13 reports the results of a regression analysis of the in-sample centrality measures and their relationship with the out-of-sample Sharpe ratio.

The first two models are directly comparable and evaluate the impact of the eigenvector centrality of the SRN (Model A) and the FMN (Model B) on the Sharpe ratio realized in a later time period. The results of the standardized regression coefficients indicate a more

Table 10. Pearson correlation coefficients (weighted networks).

| Network | Multiplex | Idiosyncratic return | Volatility | Trading volume |
|---|---|---|---|---|
| Multiplex | | | | |
| Idiosyncratic return | 0.749* | | | |
| Volatility | 0.676* | 0.355* | | |
| Trading volume | 0.683* | 0.181* | 0.201* | |
| Stock Return | 0.575* | 0.632* | 0.398* | 0.181* |

*denotes significant level of 0.1% based on the QAP with 9,300 iterations, where the probability that the observed correlation coefficient is greater than expected is 1.000.

**Table 11. Network regression results (weighted networks).**

| MODEL 1 | Dependent variable: Multiplex network | Coefficient | P (expected> = observed) | P (expected = observed) | P (expected < = observed) |
|---|---|---|---|---|---|
| | Constant | 0.001*** (0.000) | – | – | – |
| | Idiosyncratic return | 0.716*** (0.003) | 0.000 | 0.000 | 1.000 |
| | Volatility | 0.591*** (0.003) | 0.000 | 0.000 | 1.000 |
| | Trading volume | 0.781*** (0.003) | 0.000 | 0.000 | 1.000 |
| | *F-test* | 520,726.109*** | 0.000 | 0.000 | 1.000 |
| | *R²* | 0.879 | | | |
| MODEL 2 | Dependent variable: Stock return network | Coefficient | P (expected> = observed) | P (expected = observed) | P (expected < = observed) |
| | Constant | 0.001*** (0.000) | – | – | – |
| | Idiosyncratic return | 0.432 *** (0.002) | 0.000 | 0.000 | 1.000 |
| | Volatility | 0.199*** (0.003) | 0.000 | 0.000 | 1.000 |
| | Trading volume | 0.032*** (0.002) | 0.000 | 0.000 | 1.000 |
| | *F-test* | 55,638.772*** | 0.000 | 0.000 | 1.000 |
| | *R²* | 0.436 | | | |
| MODEL 3 | Dependent variable: Multiplex network | Coefficient | P (expected> = observed) | P (expected = observed) | P (expected < = observed) |
| | Constant | 0.002*** (0.000) | – | – | – |
| | Stock return network | 0.477*** (0.003) | 0.000 | 0.000 | 1.000 |
| | *F-test* | 106,606.260*** | 0.000 | 0.000 | 1.000 |
| | *R²* | 0.264 | | | |

Number of observations $N(N-1) = 215,760$; standard error for each coefficient in parentheses; MR QAP – Double-Semi-Partialing (DSP) method; number of iterations for each model: 46,500;

***Denotes significance at the 0.1% level (based on standard regression analysis).

**Table 12. Critical threshold for robustness to failure (weighted networks).**

| Network | SRN | FMN | Layer 1 | Layer 2 | Layer 3 |
|---|---|---|---|---|---|
| $f_c$ | 0.724 | 0.852 | 0.339 | 0.684 | 0.724 |

**Table 13. Regression analysis results for out-of-sample Sharpe ratio.**

| Model | Model A | Model B | Model C | Model D |
|---|---|---|---|---|
| Dependent variable | Sharpe ratio (out-of-sample) | | | |
| Eigenvector centrality_SRN | 0.114* (0.047) | | 0.030 (0.053) | -0.027 (0.057) |
| Eigenvector centrality_FMN | | 0.183*** (0.047) | 0.167** (0.053) | 0.251** (0.080) |
| Degree centrality_SRN | | | | 0.166* (0.064) |
| Degree centrality_FMN | | | | -0.175* (0.086) |
| Intercept | 0.043*** (0.003) | 0.037*** (0.003) | 0.037*** (0.003) | 0.036*** (0.005) |
| Number of observations (*n*) | 456 | 456 | 456 | 456 |
| *F*-test | 5.990* | 15.673*** | 7.989*** | 5.857*** |
| Adjusted $R^2$ | 0.011 | 0.031 | 0.030 | 0.041 |
| Max. *VIF* | 1.000 | 1.000 | 1.332 | 3.487 |

Standardized regression coefficients are provided for the independent variables; standard error in parentheses.

***, **, and *denote significant levels at 0.1%. 1%. and 5%. Respectively.

pronounced effect of the FMN than of the SRN. Furthermore, the *adjusted $R^2$* value for Model B is greater than that of Model A. Model C includes the eigenvector centrality of both networks and confirms a larger regression coefficient for the FMN. In addition, the regression coefficient for the SRN is not statistically significant. The last model (D) includes the degree and eigenvector centrality derived from both networks. The effect of the FMN centrality measures on the Sharpe ratio is more pronounced than that of the SRN centrality measures, as evidenced by the absolute values of the standardized regression coefficients. Additionally, the regression coefficient for the eigenvector centrality_SRN is not statistically significant. Note that $Max.VIF \ll 10$, indicating that there is no problem with multicollinearity.

## 5. Discussion and concluding remarks

In this paper, we have proposed the construction of a financial multiplex network, comprising three layers of the US stock market network, that considers interactions between companies. Each network layer is generated from the cross-correlation matrix of the complex financial system using the MST approach. These layers consist of i) the firm-specific stock return premium network, ii) the volatility network, and iii) the trading volume network. The analysis focuses on comparisons of the cross-correlations of the log-return network (SRN), the single-layer networks, and the financial multiplex network.

In summary, the comparison of the multiplex network, the three layers of the multiplex, and the stock return network reveals similarities and differences in their structures. The results demonstrate that the FMN has a higher density of connections and degree of connectivity, as well as a more compact structure, than the other networks. While all the networks obey a power-law vertex degree distribution, the SRN and each of the three layers exhibits a tree-like structure, a strong community structure, and disassortative behavior. In contrast, the multiplex network has none of the typical structures, has a weaker community structure with fewer modules, and has less pronounced disassortative behavior. Additionally, only the multiplex network displays the small-world property, which results from the nature of its construction. This does not imply that the MLN outperforms the SRN. These observations indicate that different networks can exhibit unique structural and behavioral characteristics and have distinct properties, highlighting the importance of understanding the underlying phenomena they represent.

From the similarity measures, one can conclude that each network layer provides a unique part of the information that contributes to the overall structure of the multiplex network. The edge overlapping between the three layers with the multiplex network is approximately 8%. Moreover, significant differences between the three layers of the multiplex network and the commonly analyzed MST-based stock return network are noticeable. The inter-layer mutual information shows that the degree distribution is not strongly correlated between the layers of the financial multiplex network. This provides the basis for an affirmative answer to the first research question: "Is there a significant difference between the FMN and the SRN?"

Furthermore, empirical studies based on the MR-QAP regression reveal that the multiplex network and the stock return network can be significantly explained by the three layers. This supports the conclusion that the distinct layers convey separate information. In other words, the results of this study indicate that the multiplex network and the stock return network are influenced by three distinct channels, with the multiplex network exhibiting a different structure. Specifically, the three network layers only explain about 50% of the variance of the stock return network, while, in an inverse relationship, the stock return network explains only approximately 26% of the variance of the multiplex network. This implies that the two independent networks are different from each other as they consider different information in the financial market. Nevertheless, about 31% of the overlapping edges in both these

networks are generated by the relatively more similar layer of the multiplex network based on the firm-specific factor of stock return to the SRN which additionally incorporates a systematic risk premium (common market factor). In other words, the idiosyncratic stock return network layer determines the stock return network to a greater extent than the other two layers of the multiplex network. In general, the low similarity reduces the risk of over-fitting and redundancy in multiplex network models. The divergence ensures that each layer contributes distinct structural and relational information, thereby enhancing the robustness and interpretability of the network. This divergence highlights the multiplex network's ability to capture the multifaceted nature of financial markets by incorporating non-redundant perspectives. It should be pointed out that the observed transitivity, higher density and much smaller fraction of pendant nodes in the FMN compared to the SRN indicate that the proposed multiplex representation has less network fragility to random removal of vertices or edges. This ultimately leads to the fragmentation of the network into two or more distinct components. Moreover, FMN is more robust to the collapse of the largest component due to the removal of randomly selected nodes. The above conclusions legitimize the affirmative answer to the second research question: "Does each layer of the stock network offer unique information within the FMN?"

The out-of-sample evaluation of the dependency structure representation indicates that the financial multiplex network has a greater impact than the stock return network on the future stock performance under the return-risk ratio in the financial market. This statement provides a clear answer to the third research question: "Which network more accurately reflects the interconnections between financial assets?" Nevertheless, the low values of the determination coefficient ( $adjusted\ R^2 < 5\%$ ) of the proposed regression models indicate the necessity for further research in this area.

This research has several limitations. Firstly, the OLS estimator was employed to extract the idiosyncratic stock returns, which may exhibit slight limitations in adequacy when applied to long time series. Secondly, the analysis may underestimate the impact of extreme events due to the overall low similarity between the volatility and turnover volume layers. This limitation arises from focusing on the entire distribution rather than explicitly isolating tail dependencies. Furthermore, such extreme events in financial markets have the potential to trigger significant structural breaks in networks. However, the adopted long study period should adequately mitigate their impact.

This study contributes to the area of complex networks in financial markets. The information space of the multiplex network is much wider than the standard range of information included in the cross-correlation of the stock return network, as the multiplex network contains three layers: firm-specific stock return, volatility, and turnover volume. Reducing data complexity by applying the MST-approach to the construction of each of the three layers of the multiplex network, on the one hand, and the three-layer bonding of a wider context of information in the financial market, on the other hand, makes the concept proposed in this study, that of extracting the backbone of the financial multiplex network, an inspiration for further research analyzing the interconnectedness of the complex financial system using the network approach. Future studies should aim to more deeply analyze the relationships between the financial multiplex network and firm performance, such as stock returns, and the volatility of the stock price. A network dynamic approach could be applied in this area. One promising direction for future research involves exploring the application of simplicial complexes to evolving multilayer networks, particularly in contexts where temporal and higher-order interactions play a crucial role. Simplicial complexes provide a means of capturing both local and global properties of networks, thereby facilitating a more profound analysis of these systems [109]. These mathematical constructs are instrumental in modeling and

analyzing complex structures within networks. Moreover, simplicial complexes extend the analytical capabilities of financial multilayer networks, enabling the simultaneous modeling of multilateral interactions both within individual layers and between them. They also facilitate the representation of interactions among multiple entities, capturing relationships that extend beyond pairwise connections. This capability is particularly advantageous in complex systems where higher-order interactions play a significant role [110,111].

## Supporting information

**S1 File. Supporting information for: The dependency structure of the financial multiplex network model: New evidence from the cross-correlation of idiosyncratic returns, volatility, and trading volume.** The Supporting Information, S1 File, contains additional tables and graphs for the robustness analysis.
(PDF)

## Author contributions

**Conceptualization:** Dariusz Siudak.

**Data curation:** Dariusz Siudak.

**Formal analysis:** Dariusz Siudak.

**Investigation:** Dariusz Siudak.

**Methodology:** Dariusz Siudak.

**Resources:** Dariusz Siudak.

**Supervision:** Dariusz Siudak.

**Validation:** Dariusz Siudak.

**Visualization:** Dariusz Siudak.

**Writing – original draft:** Dariusz Siudak.

**Writing – review & editing:** Dariusz Siudak.

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
