## [Decision Letter · Decision Letter 0]

11 Dec 2024

PONE-D-24-44663The dependency structure of the financial multiplex network modelPLOS ONE

Dear Dr. Siudak,

Thank you for submitting your manuscript to PLOS ONE. After careful consideration, we feel that it has merit but does not fully meet PLOS ONE’s publication criteria as it currently stands. Therefore, we invite you to submit a revised version of the manuscript that addresses the points raised during the review process.

In your revision you need to address fully all the comments from both reviewers. I particularly advise you to try to make your paper more targeted and emphasize what is its novelty. You may consider adjusting the title to reflect that. Furthermore, I suggest to add a more elaborated conceptualization and literature reviews suggested by one of reviewers and more critical approach to your methodology, as well as more appropriate interpretation of results. Finally, additional robustness tests suggested by one of the reviewers, would greatly enhance the contribution of your manuscript.

We look forward to receiving your revised manuscript.

Kind regards,

Srebrenka Letina, Ph.D.

Academic Editor

PLOS ONE

2. For studies involving third-party data, we encourage authors to share any data specific to their analyses that they can legally distribute. PLOS recognizes, however, that authors may be using third-party data they do not have the rights to share. When third-party data cannot be publicly shared, authors must provide all information necessary for interested researchers to apply to gain access to the data. (https://journals.plos.org/plosone/s/data-availability#loc-acceptable-data-access-restrictions) For any third-party data that the authors cannot legally distribute, they should include the following information in their Data Availability Statement upon submission: 1) A description of the data set and the third-party source 2) If applicable, verification of permission to use the data set 3) Confirmation of whether the authors received any special privileges in accessing the data that other researchers would not have 4) All necessary contact information others would need to apply to gain access to the data

Additional Editor Comments (if provided):

Reviewers' comments:

Reviewer's Responses to Questions

**Comments to the Author**

1. Is the manuscript technically sound, and do the data support the conclusions?

Reviewer #1: Yes

Reviewer #2: Partly

2. Has the statistical analysis been performed appropriately and rigorously? 

Reviewer #1: Yes

Reviewer #2: Yes

3. Have the authors made all data underlying the findings in their manuscript fully available?

Reviewer #1: Yes

Reviewer #2: Yes

4. Is the manuscript presented in an intelligible fashion and written in standard English?

Reviewer #1: Yes

Reviewer #2: Yes

5. Review Comments to the Author

Reviewer #1: I found the paper interesting. I think that this application of multiplex networks is a proper research question.

I would only recommend two things:

1- explain better in which sense (and provide much more focused literature holding the idea that) financial markets are "naturally" multiplex.

2- I think that in this paper a section devoted to the consequent application to simplicial complexes cannot be missing. At the end, although the manuscript is a nice piece to read, it still lacks "the innovative" strike. In spite of the fact that the author tries to underline the innovative nature of the work, a true innovation would be in the field of simplicial complexes, which are attracting growing attention in network theory.

Reviewer #2: Detailed comments in attachment

The manuscript proposes to understand equity markets using a multiplex with an original three-layer structure: minimal spanning tree (MST) of returns purged of the systematic component, MST of volatilities, and MST of trading volumes.\\ The author proposes a comparative investigation of the network properties of the various layers of the multiplex, of the multiplex summarized by the recurrence of edges between pairs of layers (Financial Multilateral Network, FMN), and of the classical MST of time series of returns (stock return network, SRN). In particular, he finds the properties of i) “small world” and ii) robustness to node deletion, in the case of the FMN exclusively. He also finds, again in the case of FMN, a better out-of-sample predictability, and a better ability to predict the Sharpe ratio.

All these elements are a priori interesting. However, an overall investigation of their robustness remains to be carried out, in particular to the structure of the database and its pre-processing, the choice of graph models, and the ability of the FMN to represent equity markets with a relevant sectoral structure. A systematic review in relation to benchmark portfolio theories could usefully structure this revision work.

Furthermore, the interpretation of certain results is questionable. In particular, the “small world” and node deletion robustness properties of FMN do not prove that this model represents the underlying structure of equity markets in a more relevant way than other reference graphs, in the absence of consensus on the usual verification of these properties within financial networks.

6. PLOS authors have the option to publish the peer review history of their article (what does this mean? ). If published, this will include your full peer review and any attached files.

**Do you want your identity to be public for this peer review?** For information about this choice, including consent withdrawal, please see our Privacy Policy .

Reviewer #1: No

Reviewer #2: No

---

## [Author Response · Author response to Decision Letter 1]

16 Jan 2025

Responses to Editor

Dear Editor:

Thank you very much for the referee report on my manuscript “The dependency structure of the financial multiplex network model”. The article has been revised in accordance with the recommendations set forth by the reviewers.

The revised manuscript includes:

1) restructuring the literature review and conducting an additional literature review (related to portfolio optimization),

2) corrections, extensions, and additional explanations and justifications of the adopted research methodology,

3) revisions to the interpretation of the results obtained,

4) additional robustness tests.

Furthermore, in response to your advice, the title of the manuscript has been expanded to: “The dependency structure of the financial multiplex network model: New evidence from the cross-correlation of idiosyncratic returns, volatility, and trading volume”.

Please find enclosed a detailed, point-to-point response to all Reviewer comments with all changes clearly specified. Changes in the text are marked with colors, as follows: Reviewer #1: red; Reviewer #2: blue.

I wish to take this opportunity to thank the Reviewers for their time, effort and insightful comments that have resulted in an improved manuscript. I would also like to thank you for handling my paper. Hopefully, my revised manuscript can be regarded as meeting the high standards of PlosOne journal.

Response to Reviewer #1:

I would like to thank the referee because he/she provided a considerable number of comments to improve my paper. The newly added and significantly revised portions in the paper are marked in red to make them easy to identify. Please refer to the 'Response to Reviewers' file for more details.

Response to Reviewer #2:

Thank you for your valuable feedback on my manuscript. Your insightful comments and suggestions have been invaluable in enhancing the quality of our work. Text changes are marked in blue. Details can be found in the 'Response to Reviewers' file.

---

## [Decision Letter · Decision Letter 1]

20 Feb 2025

PONE-D-24-44663R1The dependency structure of the financial multiplex network model: New evidence from the cross-correlation of idiosyncratic returns, volatility, and trading volumePLOS ONE

Dear Dr. Siudak,

Thank you for submitting your manuscript to PLOS ONE. After careful consideration, we feel that it has merit but does not fully meet PLOS ONE’s publication criteria as it currently stands. Therefore, we invite you to submit a revised version of the manuscript that addresses the points raised during the review process.

In your next submission, please address the minor comments provided by the reviewer. The editor will assess the revisions, and since these are relatively small changes, the manuscript is not expected to undergo another round of reviews.

We look forward to receiving your revised manuscript.

Kind regards,

Srebrenka Letina, Ph.D.

Academic Editor

PLOS ONE

Journal Requirements:

Reviewers' comments:

Reviewer's Responses to Questions

**Comments to the Author**

1. If the authors have adequately addressed your comments raised in a previous round of review and you feel that this manuscript is now acceptable for publication, you may indicate that here to bypass the “Comments to the Author” section, enter your conflict of interest statement in the “Confidential to Editor” section, and submit your "Accept" recommendation.

Reviewer #2: (No Response)

2. Is the manuscript technically sound, and do the data support the conclusions?

Reviewer #2: Yes

3. Has the statistical analysis been performed appropriately and rigorously? 

Reviewer #2: No

4. Have the authors made all data underlying the findings in their manuscript fully available?

Reviewer #2: Yes

5. Is the manuscript presented in an intelligible fashion and written in standard English?

Reviewer #2: Yes

6. Review Comments to the Author

Reviewer #2: All comments have been adequately addressed and justified in detail in the author's reply.

In particular, the contribution is now adequately situated, including in relation to the literature in the field of portfolio optimization, the study of the robustness of the proposed graph model is now clearly separated from the notion of efficiency of the portfolios represented, and the readability of the graphs is greatly improved by the use of node colors corresponding to the industries. This last point also makes it possible to verify that clustering by industry does indeed exist, which is an expected property of financial networks.

I simply suggest displaying on the graphs the tickers of assets exceeding a centrality threshold to be defined. For example, in figure 7, the graph has an atypical structure. A node is isolated on the right-hand side of the figure. It is characterized by both a very high degree and a very large distance from other nodes, which constitutes a highly uncommon association. It would be useful to know which asset it is to assess the reasons for this.

7. PLOS authors have the option to publish the peer review history of their article (what does this mean? ). If published, this will include your full peer review and any attached files.

**Do you want your identity to be public for this peer review?** For information about this choice, including consent withdrawal, please see our Privacy Policy .

Reviewer #2: **Yes: ** Cécile Bastidon

---

## [Author Response · Author response to Decision Letter 2]

21 Feb 2025

Response to Reviewer #2:

I would like to thank the referee because he/she provided a considerable number of comments to improve my paper. The newly added and significantly revised portions in the paper are marked in red to make them easy to identify.

Comment 1: “All comments have been adequately addressed and justified in detail in the author's reply.

In particular, the contribution is now adequately situated, including in relation to the literature in the field of portfolio optimization, the study of the robustness of the proposed graph model is now clearly separated from the notion of efficiency of the portfolios represented, and the readability of the graphs is greatly improved by the use of node colors corresponding to the industries. This last point also makes it possible to verify that clustering by industry does indeed exist, which is an expected property of financial networks.

I simply suggest displaying on the graphs the tickers of assets exceeding a centrality threshold to be defined. For example, in figure 7, the graph has an atypical structure. A node is isolated on the right-hand side of the figure. It is characterized by both a very high degree and a very large distance from other nodes, which constitutes a highly uncommon association. It would be useful to know which asset it is to assess the reasons for this.”

Reply: Thank you for the positive feedback on the addressed revisions and for the additional remark. The network structure in Figure 7 is indeed uncommon. In accordance with the suggestion, ticker labels have been added in Figure 7 for the four assets whose degree centrality (in the weighted network of the multiplex graph) exceeds the threshold (degree centrality > 0.05).

page 35, lines 777-787:

“Figure 7 illustrates the visualization of the multiplex graph (FMN) using a force-directed layout, which was generated with the ForceAtlas2 algorithm (node colors correspond to sector affiliation). It should be stressed that comparisons of the statistical properties of the networks were omitted, as weighted and unweighted networks inherently differ. The robustness test is applied to the analyses presented in Sections 4.3, 4.4, and 4.5.”

are revised into,

“Figure 7 illustrates the visualization of the multiplex graph (FMN) using a force-directed layout, which was generated with the ForceAtlas2 algorithm (node colors correspond to sector affiliation). In the presented graph, four companies (PRU, MSFT, AMP, ETN) have been highlighted, as their degree centrality in the weighted multiplex network exceeds the threshold value of 0.05 (degree centrality > 0.05). Two of these firms belong to the Financial Services sector, while one represents the Technology sector and another the Industrials sector. Notably, the AMP node stands out due to both its high degree of connectivity and its considerable distance from other vertices.

It should be stressed that comparisons of the statistical properties of the networks were omitted, as weighted and unweighted networks inherently differ. The robustness test is applied to the analyses presented in Sections 4.3, 4.4, and 4.5.”

---

## [Editor Report · Decision Letter 2]

25 Feb 2025

The dependency structure of the financial multiplex network model: New evidence from the cross-correlation of idiosyncratic returns, volatility, and trading volume

PONE-D-24-44663R2

Dear Dr. Siudak,

We’re pleased to inform you that your manuscript has been judged scientifically suitable for publication and will be formally accepted for publication once it meets all outstanding technical requirements.

Kind regards,

Srebrenka Letina, Ph.D.

Academic Editor

PLOS ONE
---

## [Editor Report · Acceptance letter]

PONE-D-24-44663R2

PLOS ONE

Dear Dr. Siudak,

I'm pleased to inform you that your manuscript has been deemed suitable for publication in PLOS ONE. Congratulations! Your manuscript is now being handed over to our production team.

Kind regards,

on behalf of

Dr. Srebrenka Letina

Academic Editor

PLOS ONE
